# Differentiating Fordyce Spots from Their Common Simulators Using Ultraviolet-Induced Fluorescence Dermatoscopy—Retrospective Study

**DOI:** 10.3390/diagnostics13050985

**Published:** 2023-03-04

**Authors:** Paweł Pietkiewicz, Cristian Navarrete-Dechent, Mohamad Goldust, Katarzyna Korecka, Verce Todorovska, Enzo Errichetti

**Affiliations:** 1Independent Researcher, 60-814 Poznań, Poland; 2Polish Dermatoscopy Group, 61-683 Poznań, Poland; 3Department of Dermatology, Escuela de Medicina, Pontificia Universidad Católica de Chile, Santiago 8331150, Chile; 4Melanoma and Skin Cancer Unit, Escuela de Medicina, Pontificia Universidad Católica de Chile, Santiago 8331150, Chile; 5Department of Dermatology, Yale School of Medicine, Yale University, New Haven, CT 06510, USA; 6Department of Skin Diseases, Regional Hospital, 60-479 Poznań, Poland; 7Independent Researcher, 1000 Skopje, North Macedonia; 8Institute of Dermatology, Santa Maria della Misericordia University Hospital, 33100 Udine, Italy

**Keywords:** ultraviolet radiation, dermoscopy, Fordyce spots, mucoscopy, fluorescence

## Abstract

Fordyce spots (FS) are heterotopic sebaceous glands affecting mostly oral and genital mucosa, commonly misdiagnosed with sexually transmitted infections. In a single-center retrospective study, we aimed to assess the ultraviolet-induced fluorescencedermatoscopy (UVFD) clues of Fordyce spots and their common clinical simulants: molluscum contagiosum, penile pearly papules, human papillomavirus warts, genital lichen planus, and genital porokeratosis. Analyzed documentation included patients’ medical records (1 September–30 October 2022) and photodocumentation, which included clinical images as well as polarized, non-polarized, and UVFD images. Twelve FS patients were included in the study group and fourteen patients in the control group. A novel and seemingly specific UVFD pattern of FS was described: regularly distributed bright dots over yellowish-greenish clods. Even though, in the majority of instances, the diagnosis of FS does not require more than naked eye examination, UVFD is a fast, easy-to-apply, and low-cost modality that can further increase the diagnostic confidence and rule out selected infectious and non-infectious differential diagnoses if added to conventional dermatoscopic diagnosis.

## 1. Introduction

Dermatoscopy is a noninvasive diagnostic technique based on the Tyndall effect and Rayleigh scattering phenomenon. The method allows for the visualization of skin structures invisible to the naked eye by the elimination of skin surface reflection. It has significantly improved diagnostic performance in skin cancer diagnostics compared to the naked-eye examination alone; it has also decreased the number needed to excise by avoiding unnecessary biopsies [1,2]. 

The idea of multispectral imaging was to observe structural differences with dermatoscopy under various wavelengths (usually from both ends of the spectrum of the visible light, namely red and blue) based on their refractive indexes difference and the differences in the absorption spectrum of particular chromophores (mainly melanin and hemoglobin). Skin refractive index for the blue light (n_B_) is higher than for the red light (n_R_). Blue light penetrates more shallow layers of the skin, yet it provides a better color contrast between melanin and blood [3]. Melanin absorbs mostly ultraviolet (UV) and blue wavelengths, thus it appears darker under this light, whereas the peak absorption of hemoglobin (both oxygenated and deoxygenated) belongs to the UV and green spectrum [3]. While dermatoscopy in UV light is based on the same concept, with an even higher refractive index (n_UV_) allowing even more shallow penetration, it also evokes “UV fluorescence” [4,5,6]. As the optical system of UV-induced fluorescence dermatoscopes filters out the reflected UV spectrum, the observer cannot see it. The visible light spectrum of light perceived by the clinician originates from the fluorochromes emitting UV-excited luminescence in a phenomenon called Stokes shift [7]. This excited luminescence (usually characterized by the longer wavelength) originates from the substances absorbing the UV electromagnetic radiation and thus achieving a higher energy level (excited state). The excited fluorochrome moves back to the ground energy level by emitting new photons, the source of the observable colors (Figure 1). Such observations have already been described in scabies and trichobacteriosis axillaris [8,9].

Fordyce spots (FS) are atrichial (not associated with follicular units) heterotopic sebaceous gland hyperplasia. FSs were described by the American dermatologist John Addison Fordyce in 1986 [10]. They correspond to small, 1–3 mm, flat-to-elevated, palpable, white-yellowish, discrete, confluent papules transilluminating through the epithelium. They are usually bilateral and symmetrically distributed and can often be found on the vermilion portion of the lips, oral jugal mucosa (commonly buccal or retromolar area), and genital mucosa in both sexes (glans, foreskin, labia), but may infrequently also affect other sites (“atypical sites”), including areola and esophageal mucosa [11,12,13,14,15,16,17,18,19,20]. Primordial ectodermal tissue involved in maxilla/mandible formation is regarded to be the source of FS formation [21]. There is an inconsistency in sex prevalence reported—some studies note male predominance, whereas others equal gender distribution [13,18,22,23]. FSs are estimated to affect 80–90% of the population; however, they are likely underreported due to being asymptomatic [24]. Even though present at birth, in many instances FSs are perceived by the patients as new and are commonly suspected to be viral warts, herpes vesicles, or other sexually transmitted disease, either by the patient himself or the physician [25]. Therefore, they are an important complaint in clinics. 

In the majority of cases, patients seek help in early adulthood, when the gland enlargement is stimulated by the hormonal factors in puberty. For this reason, FSs can cause high anxiety and somatic symptom disorder [26]. Nevertheless, FSs have also been reported in neonates (1%) due to maternal hormone activity [27,28,29]. Although the majority of cases are harmless and do not require any treatment (unless for purely cosmetic reasons), reports on possible neuro-sebaceous associations, cardiovascular risk, and sebaceous glands system activation in hereditary non-polyposis colorectal cancer syndrome (i.e., Lynch syndrome Muir-Torre variant, although with preserved mismatch repair protein expression in FS) can be found in the literature [13,30,31,32,33]. Possible modalities for the treatment of FS include carbon dioxide laser or high-power diode laser ablation [20], electrocautery and curettage [34], micro-punching [35], peelings [36], photodynamic therapy with 5-aminolevulinic acid [37,38], intense pulse light or blue light with 5-aminolevulinic acid [39], single insulated microneedle radiofrequency device [40,41], isotretinoin treatment [42,43], or combined therapies [44].

Commonly, the diagnosis of FS can be easily performed clinically, yet dermatoscopic examination can be useful in more challenging cases or those with atypical presentation [45]. To the best of our knowledge, FS UVFD features have never been reported in the literature. The aim of this study was to assess the UVFD clues of Fordyce spots and their common clinical simulants.

## 2. Materials and Methods

This was a retrospective study performed at a single dermatology center. Dermatology Private Practice patient’s records (1 September–30 October 2022) were reviewed for the diagnoses of FS, molluscum contagiosum (MC), penile pearly papules (PPP), human papillomavirus (HPV) warts, genital lichen planus (LP), and genital porokeratosis. After obtaining the patients’ consent, anonymized important data concerning the age, sex, cause for the presentation, lesion location, and photodocumentation, were collected. Photodocumentation included clinical images as well as polarized, non-polarized, and UVF (measured peak wavelength 375 nm) dermatoscopy images (Xiaomi Mi 10T Pro 5G, Xiaomi, Beijing, China, paired with DL5, Dermlite, San Juan Capistrano, California, USA; ×10 magnification, with water used as an immersion medium in genital and oral mucosa).

A total of 12 FS patients (1 female, 11 males) and 14 patients in the control group (2 females, 12 males) representing common FS clinical differentials were included in the study. Control group diagnoses included genital lichen planus (2 cases, 2 lesion captured), molluscum contagiosum (3 cases, 8 lesions captured), penile pearly papules (3 cases, 3 lesions captured), genital porokeratosis (1 case, 1 lesion captured), and human papillomavirus warts (5 cases, 6 lesions captured). Due to the benign nature of the lesions and confident clinical-dermatoscopic diagnosis, none of them were biopsied. Dermatoscopic clues were analyzed in line with the latest International Dermoscopy Society consensus on inflammatory dermatoses [46].

## 3. Results

A total of 25 patients were included (12 cases, 14 controls). The mean age for the FS group was 33.1 years (SD 5.3, min. 23, max. 43), and 32.3 years for the control group (SD 13.1, min. 6, max. 47) (t-Student test, unpaired, *p* = 0.85). Clinical and demographic characteristics of FS and control groups are presented in Table 1 and Table 2, respectively.

In the FS group, five out of twelve (41.7%) patients attended their visits suspecting a sexually transmitted infection, whereas in the remaining seven cases (58.3%) it was an incidental finding during routine examination. For the control group these numbers were nine (64.3%) and five (35.7%), respectively. No significant differences between the groups were noted in regard to suspected cause (Chi-square test, *p* = 0.249). The most common sites affected in the FS group were upper lips (50%), followed by the foreskin (33.3%), and glans (16.7%). For the control group, it was pubis (30%), glans (25%), neck and penile shaft (10% each), foreskin, popliteal fossa, areola, upper leg, and scalp (5% each).

In polarized dermatoscopy, FSs were seen as well or poorly demarcated (more superficial and deep-seated lesions, respectively), clustered, roundish white-yellowish clods. Central, slightly brighter dots marking the opening of sebaceous glands were incidentally seen in a minority of images (two oral lesions, both with well-demarcated FS). UVFD in 11 out of 12 (91.7%) FS cases displayed predominantly bright blue/green fluorescent dots within the UVRD-neutral clods, corresponding to the sebaceous gland duct openings and the glands, respectively (Figure 2). In one FS case (8.3%) all the dots within the UVFD-neutral clods were blue/red. We have also found single brighter dots to be yellow, orange or red in some of the images. We did not visualize these findings in any of the UVFD images in the control group (Fisher’s exact test, two-sided; *p* < 0.001).

Non-polarizing-specific white structures of LP turned into poorly demarcated UVFD-dark lines, whereas polarizing-specific white lines were not visible in UVFD (Figure 3). Erythematous areas seen at the periphery of the LP lesions in polarized dermatoscopy proved to be seemingly larger and darker in UVFD.

Polarized dermatoscopy of MCV papules showed central yellow-white clod/grouped clods enclosed in single or clustered skin-colored clods surrounded by the linear vessels arranged radially, sparing the central aspect of the lesion. At the central portion of each papule there was a whitish pore/pores, sometimes surrounded with a white circle, corresponding to clinically visible umbilication. In UVFD, the papules were dark with no white circles, and some pores featured faded yellowish fluorescence (Figure 4).

Polarized dermatoscopy of genital human papillomavirus warts displayed either regularly distributed dotted or/and glomerular vessels, and occasionally intersecting polarizing-specific white lines. These were located over a tan or pink structureless area. Nongenital warts showed skin-colored elongated clods, some with centered linear looped vessels and scale. The warts were either UVFD-neutral (nonpigmented lesions) or dark (pigmented lesions). Non-genital lesions featured perivascular bright halos or fluorescence of the tips of the elongated papillae. Both features were generally absent in some of the genital lesions. One case showed mild reddish/coral luminescence (Figure 5).

PPP, seen as whitish clods with centered glomerular vessels in polarized dermatoscopy, were neutral in UVFD (Figure 6).

Polarized dermatoscopy of a single non-pigmented lesion of genital porokeratosis showed a specific dermatoscopic clue, namely the annular keratotic rim (cornoid lamella), which in our case was perifollicular and non-interrupted, whereas the affected area was skin-colored. UVFD further enhanced the presence of this rim of scale and central perifollicular hyperkeratosis (UVFD-bright fluorescence), whereas the affected area was also brighter than normal skin (Figure 7).

## 4. Discussion

Herein, we have described the features of FS with UVFD compared to its clinical mimickers. In recent years, new dermatoscopes/dermatocameras have introduced UV LEDs, including the DZ-D100 (Casio, Tokio, Japan), transforming the near-UV 405 nm images into the grayscale in UV reflectance photography and DL5 (Dermlite, San Juan Capistrano, USA), the first commercially available hybrid dermatoscope with non-polarized, polarized, and UVFD modes. Even though the primary aim of UV dermatoscopy was enabling better visualization of melanoma margins, it has become clear that UV not only highlights already known structures, but also opens a box containing new set of findings, just as polarized dermatoscopy did two decades ago [47]. Currently, we can expect a plethora of new clues reported, and among them also some artifacts and false positives. 

In dermatoscopy, wavelength-dependent differences in refractive indexes are responsible for the chromatic dispersion. The light emitted by the dermatoscope diodes refracts into separate colors when it passes through the epidermis. Longer wavelengths, close to red (620–750 nm), have low refractive indexes and pass through the epidermis at a wider angle, illuminating and transluminating the deeper structures, whereas shorter ones, close to blue (450–495 nm), have high refractive indexes and penetrate relatively more shallowly into the dermis, enabling better visualization of the superficial structures. The refractive index for UV wavelengths (100–400 nm) is even higher than for blue ones. High energy, short UV waves, potentially dangerous to the human retina, have a shallow penetration and are easily dispersed. 

Human vision is regarded to be trichromatic. Short (S), medium (M), and long (L) cone receptors of the retina are spectral-sensitive, depending on the spectral absorption of the contained opsin type (SWS1 and LWS). S cones have peak sensitivity at 420 nm, M at 534 nm, and L at 564 nm (near-UV/violet/blue, yellow/green, and yellow/red spectra, respectively) [48]. It was a common belief that humans are unable to see UV light. A recent study by Hammond Jr. at al. showed that UV light can be detected by most individuals at a peak wavelength of 315 nm [49]. Nevertheless, the UV signal emitted by the UVFD is low and is further blocked by the optical system, so the method is not a true UV imaging technique (but rather Plato’s “reflections on the walls’’, as it does not allow reflected UV to be recorded) but the pictures obtained with the UV radiation are indeed exclusively generated with excited fluorescence due to Stokes shift [7].

Fordyce spots are heterotopic sebaceous glands, commonly occurring on oral and genital mucosa. The lesions are commonly taken for penile pearly papules, genital warts, molluscum contagiosum or other sexually transmitted infections [25]. 

The mean age of our FS patients as well as a rate of asymptomatic cases was similar to the one reported previously in the Korean study [23]. All FS lesions photographed in UVFD displayed specific bright fluorescent dots within the clods, corresponding to the sebaceous gland duct openings. Excited yellow luminescence is likely caused by the sebum in the duct ostia [50], whereas orange or red dots are possibly caused by duct colonization with porphyrin-producing commensal skin bacteria, as in case No 11 (Figure 2i) where Corynebacterium glucuronolyticum, a fermenting lipophylic Gram-positive bacteria, was isolated from the glans (+++) [51]. Protoporphyrin IX and coproporphyrin III are the fluorochromes reported to evoke orange to coral red luminescence in UV (e.g., in acne, erythrasma, or porphyria) [50]. Collagen and elastin fibers are likely the source of blue background fluorescence [52].

In contrast to FS, LP, which is the first of the FS differential diagnoses, features a lichenoid morphological pattern in pathology (consisting of band-like lymphocytic infiltrate below the dermal–epidermal junction). We deduce that this dermal infiltrate can behave as a light absorber and is likely responsible for generating the dark structureless areas seen under UVFD. MCV papules commonly feature central single or multiple white-yellowish clods and linear serpentine vessels arranged radially, sparing the central aspect of the tumor. However, MCV stroma is not visibly translucent in polarized dermatoscopy. In general, these proved to be UVFD-dark apart from central pores featuring abnormal keratinization (mollusk bodies, hyperkeratosis), likely responsible for UVFD’s faded yellowish glow. HPV warts usually present as brown or skin-colored clods/areas featuring centered vessels (dotted or linear looped, in flat and elevated lesions, respectively), white perivascular halo (a hallmark of keratinizing tumors), white polarizing-specific lines (stromal collagen alteration), and scaling (hyperkeratosis, sometimes limited to the tips of papillomatous projections, and usually absent in genital lesions). In non-genital areas, the presence of melanin (not an active UVFD fluorochrome) caused the UVFD-darkening of pigmented lesions, whereas the lack of melanin at extragenital sites made the warts UVFD-neutral. We suspect that marked hyperkeratosis is responsible for the excited bright blue fluorescence observed at the tips of the papillomatous projections in extragenital sites, as well as around the clods/centered vessels in both genital and extragenital sites. A humid environment at the genital site might be the cause of the lack of this prominent illumination in genital warts. Occasional red fluorescence observed in one case in wart furrows might have developed due to bacterial colonization of these spaces or can merely be an artifact. As no similar observations were made in regard to this red/coral glow in genital warts, this issue requires further confirmation in larger datasets and investigation of the possible cause (e.g., with microbiologic culture). Dermatoscopically pale whitish clods of PPP were UVFD-neutral, which made the lesions easily discernible from FSs. Genital porokeratosis is a rare entity with less than 20 cases reported in the literature. It tends to be misdiagnosed and mistreated as lichen planus, genital warts, or candidal or circinate balanitis for many months and years. The annular keratotic rim of scale (corresponding to parakeratotic column on histopathology) is a dermatoscopic hallmark of this heterogeneous group of diseases and a unique clue to the diagnosis [53,54]. The central area can be pink, brown, or white in polarized dermatoscopy. To date, no UVFD clues to porokeratosis have been described. We have observed a slight depigmentation in the area affected, making it UVFD-brighter (which could develop due to mild epidermal remodeling or pigment incontinence not observed in polarized dermatoscopy), and the linear ring of scale was expectedly bright blue (likely due to the presence of hyper- and parakeratosis). 

FS is one of the entities commonly mistaken for STI [55]. Young adults, especially those who are sexually active, may develop “venereophobia”. This disorder has a significant impact on human mental and sexual health, but can also be burdensome to healthcare systems, resulting in multiple supernumerary visits and sometimes administration of unnecessary treatments. Additionally, it can compromise the patient–physician relationship. Despite clinicians’ diagnosis, as the patient can clearly see the lesions, he/she might still suffer from irrational fears about the presence of infection and seek help by non-medical pseudo-therapists. It has been observed that the use of dermatoscopy has a positive impact on a patient’s decision making in the context of a diagnostic biopsy as the clinician can refer to the image to show the clues to the condition and select the best site for a biopsy [56]. In our experience, STI suspects are usually very eager to understand the condition they have and the mechanism in which it develops. We believe that UVFD imaging, showing specific bright fluorescence of the openings of sebaceous gland ducts, might play an additional supportive role to classical polarized dermatoscopy in convincing the patient about the origin of the lesion.

There are several limitations of the study. The study covers retrospective data from a single Central European site, with all the patients being Caucasian. Further, prospective studies including non-Caucasian populations should shed more light on the UVFD presentation of common genital dermatoses in a skin of color. Our research group and control groups are innumerable; thus, the distribution of patterns might be different in a real-life setting. Nevertheless, this study remains the largest study on the dermatoscopy of FS published.

Further studies on the exact chemical compounds responsible in UVFD luminescence and on the application of UVFD in other neoplastic and non-neoplastic dermatoses may contribute to the wider use of this novel method.

## 5. Conclusions

We present a novel and seemingly specific UVFD pattern of FS—regularly distributed bright dots over yellowish-greenish clods. Even though, in the majority of instances, the diagnosis of FS does not require more than naked-eye examination, UVFD is a fast, easy-to-apply, and low-cost modality that can further increase the diagnostic confidence and rule out selected infectious and non-infectious differential diagnoses if added to dermatoscopic diagnosis [57]. This simple method can promptly rule out sexually transmitted origin of the lesions, limiting unnecessary laboratory workup and avoiding unintended induction of somatic symptom disorder associated with STI resulting in repetitive unnecessary treatment and frequent medical consultations. Moreover, UVFD imaging can be used as an additive supportive tool to strengthen the patient’s belief in the benign nature of the suspect lesions.

## Figures and Tables

**Figure 1 diagnostics-13-00985-f001:**
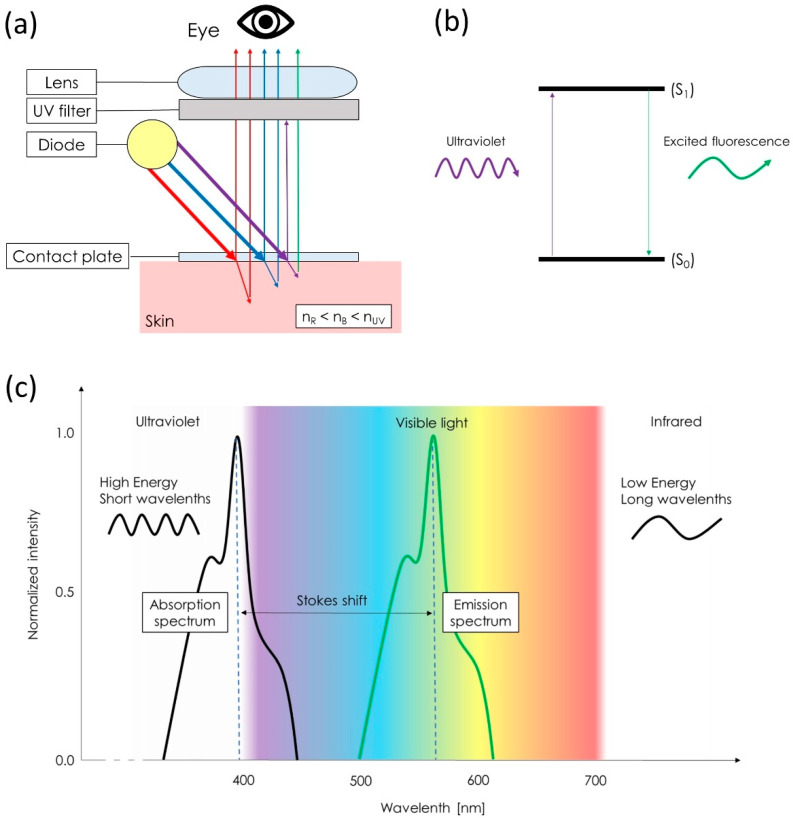
Basics of multispectral and ultraviolet-induced fluorescence dermatoscopy: Each diode of a dermatoscope emits light characterized by particular energy and wavelength that impacts its refraction in the skin, its penetration, absorption by the chromophores, scattering, and reflection. Reflected light is recorded by the observer. (**a**) Reflected UV light is blocked by the UV filter that allows the passage of the visible spectrum of light. (**b**) UV-excited skin chromophore moves from the ground vibrational level (S_0_) to the higher vibrational level (S_1_). Moving back to the ground vibrational level is accompanied by the emission of new photon (usually of a longer wavelength, so belonging to the visible spectrum of light). (**c**) This photon is responsible for the UV-excited fluorescence (Stokes shift phenomenon) seen by the observer.

**Figure 2 diagnostics-13-00985-f002:**
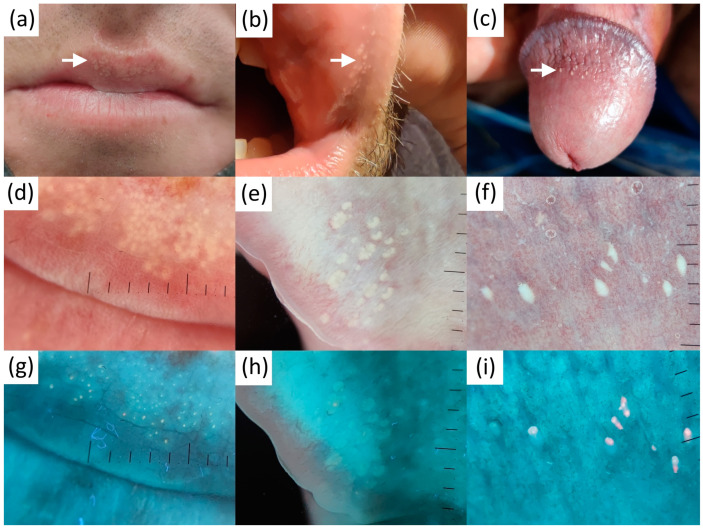
Clinical (**a**–**c**), contact polarized (**d**–**f**) and ultraviolet-induced fluorescence (**g**–**i**) dermatoscopic images of 3 representative patients with Fordyce spots.

**Figure 3 diagnostics-13-00985-f003:**
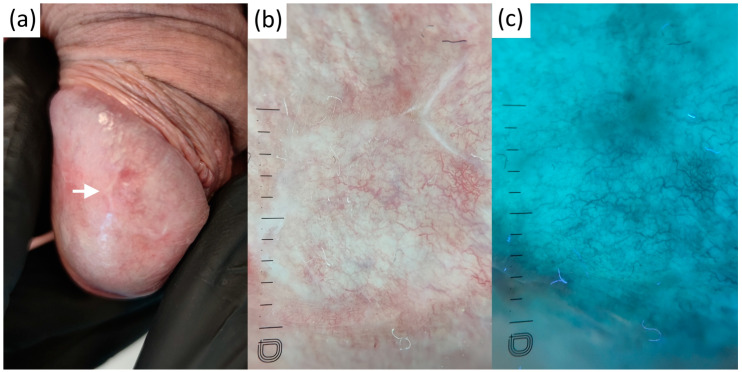
Clinical (**a**), contact-polarized (**b**), and ultraviolet-induced fluorescence (**c**) dermatoscopic images of genital lichen planus.

**Figure 4 diagnostics-13-00985-f004:**
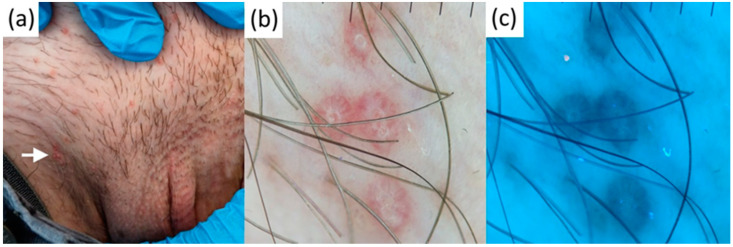
Clinical (**a**), contact-polarized (**b**), and ultraviolet-induced fluorescence (**c**) dermatoscopic images of genital molluscum contagiosum.

**Figure 5 diagnostics-13-00985-f005:**
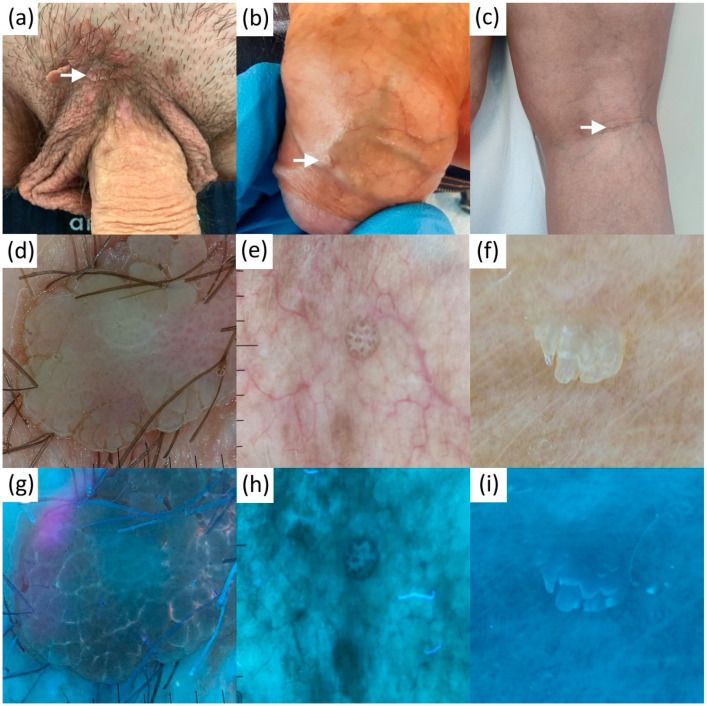
Clinical (**a**–**c**), contact-polarized (**d**–**f**), and ultraviolet-induced fluorescence (**g**–**i**) dermatoscopic images of 3 patients with genital human papillomavirus warts. Note that the violet glow in the top left portion of the photo (**g**) is an artifact.

**Figure 6 diagnostics-13-00985-f006:**
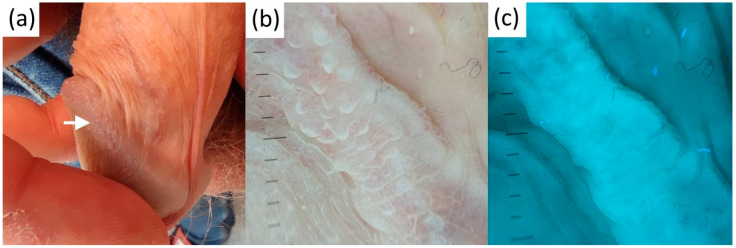
Clinical (**a**), contact-polarized (**b**), and ultraviolet-induced fluorescence (**c**) dermatoscopic images of penile pearly papules.

**Figure 7 diagnostics-13-00985-f007:**
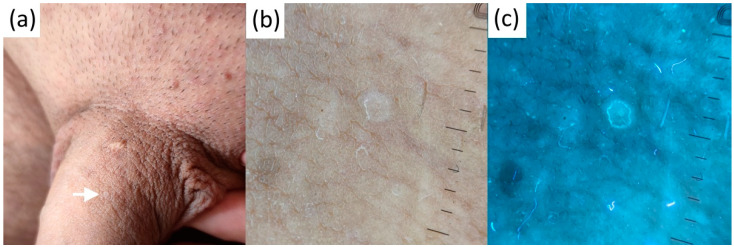
Clinical (**a**), contact-polarized (**b**), and ultraviolet-induced fluorescence (**c**) dermatoscopic images of genital porokeratosis.

**Table 1 diagnostics-13-00985-t001:** Clinical and demographic characteristics of patients with Fordyce spots.

No	Age	Sex	Ethnicity	Site	Clinical Presentation of Fordyce Spots	Context
1	33	M	Caucasian	Upper lip	Flat, poorly defined yellow-white papules grouped along the vermilion	STI suspect
2	38	M	Caucasian	Upper lip	Palpable, elevated well-defined yellowish papules grouped bilaterally in the vicinity of labial commissure of the mouth	Incidental finding
3	33	M	Caucasian	Upper lip	Slightly elevated, well-defined yellow-white papules grouped above the tubercle of the lip	STI suspect
4	23	M	Caucasian	Foreskin	Slightly elevated, well-defined yellow-white papules grouped bilaterally in the proximity of frenulum	STI suspect
5	35	M	Caucasian	Upper lip	Flat, poorly defined yellow-white papules grouped along the vermilion	Incidental finding
6	28	M	Caucasian	Foreskin	Slightly elevated, poorly defined scant yellow-white papules located bilaterally in the proximity of frenulum	Incidental finding
7	30	M	Caucasian	Foreskin	Slightly elevated, poorly defined scant yellow-white papules located bilaterally in the proximity of frenulum	Incidental finding
8	43	M	Caucasian	Upper lip	Flat, poorly defined yellow-white papules grouped unilaterally in the vicinity of left labial commissure of the mouth	Incidental finding
9	40	M	Caucasian	Foreskin	Slightly elevated, poorly defined scant yellow-white papules located bilaterally in the proximity of frenulum	STI suspect
10	29	M	Caucasian	Glans	Slightly elevated, poorly defined scant yellow-white papules clustered on the left dorsal surface	Incidental finding
11	31	M	Caucasian	Glans	Elevated, well-defined yellow-white papules clustered on the mid-dorsal surface	STI suspect
12	34	F	Caucasian	Upper lip	Elevated, well-defined yellow-white papules grouped along the vermilion	Incidental finding

Legend: F—female, No—patient’s number, M—male, STI—sexually transmitted infection.

**Table 2 diagnostics-13-00985-t002:** Clinical characteristics of the control group.

No	Age	Sex	Ethnicity	Site	Diagnosis/Clinical Presentation	Context
1	47	M	Caucasian	Glans	Genital lichen planus/slightly elevated whitepapules and annular lesions on the glans	STI suspect
2	23	M	Caucasian	Glans	Genital lichen planus/coalescing flat-top papules on the glans and foreskin	STI suspect
3	9	F	Caucasian	Pubis	Molluscum contagiosum/solitary umbilicatedpapule	Incidental finding
4.1	6	M	Caucasian	Neck	Molluscum contagiosum/multiple disseminated umbilicated papules	Incidental finding
4.2	6	M	Caucasian	Neck	Molluscum contagiosum/multiple disseminated umbilicated papules	Incidental finding
4.3	6	M	Caucasian	Scalp	Molluscum contagiosum/solitary umbilicatedpapule	Incidental finding
5.1	41	M	Caucasian	Pubis	Molluscum contagiosum/multiple disseminated and grouped umbilicated papules	STI suspect
5.2	41	M	Caucasian	Pubis	Molluscum contagiosum/multiple disseminated and grouped umbilicated papules	STI suspect
5.3	41	M	Caucasian	Pubis	Molluscum contagiosum/multiple disseminated and grouped umbilicated papules	STI suspect
5.4	41	M	Caucasian	Pubis	Molluscum contagiosum/multiple disseminated and grouped umbilicated papules	STI suspect
6	28	M	Caucasian	Glans	Penile pearly papules/multiple white papuleslocated on the corona of the penis	STI suspect
7	43	M	Caucasian	Glans	Penile pearly papules/multiple white papuleslocated on the corona of the penis	Incidental finding
8	31	M	Caucasian	Glans	Penile pearly papules/multiple white papuleslocated on the corona of the penis	STI suspect
9.1	45	F	Caucasian	Popliteal fossa	Viral wart/solitary hyperkeratotic skin-colored papule	Incidental finding
9.2	45	F	Caucasian	Upper leg	Viral wart/solitary hyperkeratotic skin-colored papule	Incidental finding
10	21	M	Caucasian	Areola	Viral wart/solitary hyperkeratotic skin-colored papule	Incidental finding
11	47	M	Caucasian	Foreskin	Viral wart/solitary skin-colored papule	STI suspect
12	35	M	Caucasian	Penile shaft	Viral wart/multiple disseminated skin-colored papules	STI suspect
13	44	M	Caucasian	Pubis	Viral wart/multiple disseminated skin-colored papules	STI suspect
14	32	M	Caucasian	Penile shaft	Genital porokeratosis/disseminated scaly papules in the genital area	STI suspect

Legend: F—female, No—patient’s number (additional image number was added if many images were captured), M—male, STI—sexually transmitted infection.

## Data Availability

Not applicable.

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
