# Peer review of "Differentiating Fordyce Spots from Their Common Simulators Using Ultraviolet-Induced Fluorescence Dermatoscopy—Retrospective Study"

_diagnostics, 2023, doi:10.3390/diagnostics13050985_

Round 1
Reviewer 1 Report
Please see the attached file.

Author Response
Review for DIAGNOSTICS-2183560
Title: Differentiating Fordyce spots from its common simulators using ultraviolet reflectance dermatoscopy - retrospective study.
In this retrospective, single-center study the authors described specific signs of Fordyce-spots (FS) with the ultraviolet reflectance dermatoscopy (UVRD) technique. The subject is interesting, however, the usefulness of the study, as focusing on the FS is debatable. As the authors also mentioned, diagnosing Fordyce spots can be quite easy with the naked eye, only in exceptional and rare cases it can cause a difficulty of differential diagnosis. Nevertheless, they also describe important, and seemingly specific UVRD signs of other diseases mainly in the genital region. Differentiating between those can be a more remarkable problem, as lichen planus and warts, especially condyloma plana, can mimic each other easily. Therefore, if the focus is modified to show UVRD as a technique to make differential diagnose among such lesions would increase the value and the clinical usefulness of the manuscript.
Thank you for this comment. We agree that in majority of cases examination other than clinical is not required, unless the lesions are flat and clinically bland. Nevertheless, many patients consulted by us over the years report a history of treatment with topical caustic products, and topical/systemic antivirals/antifungals prescribed by general practitioners, but surprisingly also by dermatologists and urologists. We suppose that flat lesions of deep-seated FS may not be easily identified at a first glance and could be misdiagnosed if clinical examination is conducted in superficial way. This results in patient’s dissatisfaction with medical service, and with ineffective treatment can inspire venerophobia, sexual disengagement or abstinence. Even though this is not a topic raised in our paper, we observe dermatoscopy-driven decision making in patients who were showed dermatoscopic images. With fluorescent UVRD images this psychological aspect of this phenomenon might be even stronger, calming down the patient and assuring him/her that the clinician is adamant about the benign nature of the lesions. We believe that our paper can even be used as an educational/psychological aid to compare patients’s own images with our findings.
In addition, there are several confusing structural issues, listed below, mainly in the Figures. Not only is the labeling confusing, but comparing FS in the typical site (glans, foreskin, etc.) with another condition located in a different anatomical site (corpus penis, mons pubis) can be inaccurate. Comparing warts (or other diseases) and FS in the same anatomical location would be more accurate.
We address this issue raised by the reviewer in the last comment.
It can also be interesting to perform a short study, including dermatologists, and other physicians/nurses/medical students to try to diagnose the lesions based on the pictures, and to compare them. This can further strengthen the usefulness of the study.
We wish to kindly thank the reviewer for this idea. We will surely use this concept in a future study when UVRD imaging will be more common, and the source of particular structures and colours better understood (currently just 13 papers published in the past 6 years).
This technique requires training and standardization of terminology as no interobserver concordance can be assessed if the observers do not understand what they look at. Moreover, this task cannot be achieved in 10day period required by the Publisher for the revision.
Besides these major and minor comments the manuscript is outstanding and well-written. The subject is significant and forward-looking, thus in conclusion, the review should be accepted for publication after correcting those above and below-listed issues.
We wish to thank the reviewer for this kind remark.
Issues:
- Fig. 2: the pictures themselves should contain “a-i” markers, as it is confusing in the present form. It is also applicable to Fig 3.-7.
The labelling of the images is corrected.
- Fig. 6. is mislabelled as Fig. 4.
The labelling was corrected.
- On clinical pictures, small arrows could help to identify particular lesions.
We have added markers to the photographed sites.
- Show warts and other differential diagnoses in the same anatomical sites as FS or in locations where FS commonly occurs.
Each example of FS differential for genital site (pubis, shaft, foreskin, glans) was included. As there can be some regional variations in presentation, especially in viral warts, we decided to broaden the scope to show different UVRD aspects.

Reviewer 2 Report
Suggestion and Recommendation:
1. In the introduction, the scientific problem of the existing evaluation is missing. It should be elucidated clearly.
2. At the end of the introduction, it is recommended to clearly state the research goal/objectives of this study and what the authors have done to address the identified research problem/section description. Remove extra description which is not relevant to your proposed system
3. Please improve writing of the paper, there are several grammatical errors and typos issues in the complete manuscript.
4. The presented diagram should be modified, try to define more. For example, explain sequences of data movement/executions in Figures such as Figure 1 and 2. Till now, it is very complex to analyze.
5. I suggest you explore more open research issues in this domain and add at least 4-6 open areas that need experts’ consideration.
6. Reference format must be uniform.
Author Response
- In the introduction, the scientific problem of the existing evaluation is missing. It should be elucidated clearly.
The aim of the study is introduced in lines 104-108 of introduction. We do not understand what Reviewer 2 in context of “scientific problem of existing evaluation”.
- At the end of the introduction, it is recommended to clearly state the research goal/objectives of this study and what the authors have done to address the identified research problem/section description. Remove extra description which is not relevant to your proposed system
The research goal is introduced in lines 104-108, whereas the methods used (character of the study, time frame of the study, data collection method and obtaining of patient’s consent, target group and control groups, all metadata analysed, as well as anonymization method are clearly presented in Materials and Methods section.
We believe that the study requires substantial introduction focusing on the disease (pathogenesis, epidemiology, associations) and its social burden, as well as introducing physical rudiments of generated images and source of the colours seen with UVRD, as the method is novel.
We do not understand what the “proposed system” is. This comment is too general to be precisely addressed.
- Please improve writing of the paper, there are several grammatical errors and typos issues in the complete manuscript.
We have proofread the paper. The Table 2 and Figure 6 labelling were corrected. We have also made minor corrections in both tables. We did not find any grammatical errors. Thus, we would be grateful if the Reviewer 2 could point out specific sections/lines where the problem occurs.
- The presented diagram should be modified, try to define more. For example, explain sequences of data movement/executions in Figures such as Figure 1 and 2. Till now, it is very complex to analyze.
We do not understand this suggestion as it is too general. We did not use any data movement/execution in Figure 1 and 2. Figure 1 is a demonstration of Stoke’s shift phenomenon and the differences between classical and UVR dermatoscopy. Figure 2 is a case presentation and includes just a visual description.
- I suggest you explore more open research issues in this domain and add at least 4-6 open areas that need experts’ consideration.
We do not understand what Reviewer 2 had in mind in regard to “open research issues” or “open research areas”. There is no existing research on UVRD in Fordyce spots. In fact, there are no more than 13 papers on UVRD published in the past 5 years.
- Reference format must be uniform.
All the references are Vancouver style as required by the publisher. As these were already uniform and in line with Diagnostics authors’ guidelines we did not make any corrections.
Reviewer 3 Report
The type of study seems to me to be of great importance since it presents a novel option for the diagnosis of FS, but I believe that 12 patients is not enough to give definitive conclusions and to be published as an international reference, it seems to me to be very poor, I would recommend that for it to be published more patients be evaluated within a minimum time range of 6 months, this would give more validity to the study. It is important that researchers with this type of study, in addition to the statistical tests they used, investigate and apply the propensity score which is being used in a practical way for this type of comparison.
Author Response
The type of study seems to me to be of great importance since it presents a novel option for the diagnosis of FS, but I believe that 12 patients is not enough to give definitive conclusions and to be published as an international reference, it seems to me to be very poor, I would recommend that for it to be published more patients be evaluated within a minimum time range of 6 months, this would give more validity to the study.
We appreciate this comment. We wish to defend our design of the study. Unfortunately, we believe we are not able to make any amendments to the study design in the fixed 10 day period required to address the review.
Even though there are 12 FS patients, the whole study group is 25 patients with a total of 32 lesions. As FS presents a consistent and unique UVRD pattern in every case, not observed in any other differential diagnosis, we decided to include this number of cases which belong to a larger international multicentre dataset on neoplastic and non-neoplastic skin disorders.
The classical dermatoscope has been used since 1922 (https://www.ncbi.nlm.nih.gov/pmc/articles/PMC8208256/). Nevertheless, there are only 2 reports in the literature on dermatoscopic presentation of FS:
1 – report on 2 cases: https://www.ncbi.nlm.nih.gov/pmc/articles/PMC6615395/
2 – 4 cases collected in two year period (March 2019 - February 2021) https://www.ncbi.nlm.nih.gov/pmc/articles/PMC9549566/
No more than 13 reports on the use of UVRD exist in the PubMed database until this day.
To present the scale of our database we wish to underline that UV reflectance dermatoscope is commercially available for less than 4 months. Thus, our collected cases constitute not only the largest UVRD database existing, but also the largest dermatoscopic series ever reported in FS.
It is important that researchers with this type of study, in addition to the statistical tests they used, investigate and apply the propensity score which is being used in a practical way for this type of comparison.
We are not acknowledged with the use of propensity score in observational studies based on dermatoscopy. We did not find any article using this score in the context of dermatoscopy (7900 papers) or pattern analysis. Our study does not focus on the outcome, but rather on describing the optical dermatoscopic-UVRD correlations. Moreover we cannot assess the interobserver concordance and how the UVRD affects the diagnostic accuracy, as there are no other datasets allowing teaching the evaluators and there are no other reports on UVRD of FS.
Reviewer 4 Report
This authors assessed the ultraviolet reflectance dermatoscopy (UVRD) clues of Fordyce spots and their common clinical simulants: molluscum contagiosum, penile pearly papules, human papillomavirus warts, genital lichen planus, and genital porokeratosis. The authors analyzed 12 FS patients and 14 patients in the control group. Results show that UVRD is a fast, easy to apply, and low-cost modality that can further increase the diagnostic confidence and rule out selected infectious and non-infectious differential diagnoses if added to traditional dermatoscopic diagnosis.
Major comments:
1. In Table 2, the clinical characteristics of the control group, there are more young age patients, will it add bias to the analysis results?
2. Some patients in Table 2 have more than one records. Please add description to those points.
Minor comments:
1. Incorrect index of Table 1 and Figure 6
Author Response
This authors assessed the ultraviolet reflectance dermatoscopy (UVRD) clues of Fordyce spots and their common clinical simulants: molluscum contagiosum, penile pearly papules, human papillomavirus warts, genital lichen planus, and genital porokeratosis. The authors analyzed 12 FS patients and 14 patients in the control group. Results show that UVRD is a fast, easy to apply, and low-cost modality that can further increase the diagnostic confidence and rule out selected infectious and non-infectious differential diagnoses if added to traditional dermatoscopic diagnosis.
Major comments:
- In Table 2, the clinical characteristics of the control group, there are more young age patients, will it add bias to the analysis results?
We agree that two patients in the control group (Patient 3 and 4) were younger than any other in the FS group. Nevertheless, we have assessed this problem and did not find any significant differences between the groups in regard to age (line 130-132: The mean age for the FS group was 33.1 years [SD 5.3, min. 23, max. 43], and 32.3 years for the control group [SD 13.1, min. 6, max. 4](t-Student test, unpaired, p=0.85]).
Even though the literature data indicates that FS spots are reported mainly in adolescents and young adults, there are reports on children and neonates, so we do not see including these patients incorrect, especially that the presentation of the lesions they displayed does not differ with age.
- Some patients in Table 2 have more than one records. Please add description to those points.
As our intention was to underline that multiple lesions (possibly with different dermatoscopic or UVRD patterns) occur in particular patients we decided to add an extra digit. This particular labelling was matched with Table’s legend: No - patient’s number (additional image number was added if many images were captured) and data in materials and Methods section – lines 120-125: A total of 12 FS patients (1 female, 11 males) and 14 patients in the control group (2 females, 12 males) representing common FS clinical differentials were included in the study. Control group diagnoses included: genital lichen planus (2 cases, 2 lesion captured), molluscum contagiosum (3 cases, 8 lesions captured), penile pearly papules (3 cases, 3 lesions captured), genital porokeratosis (1 case, 1 lesion captured), and human papillomavirus warts (5 cases, 6 lesions captured).
Minor comments:
- Incorrect index of Table 1 and Figure 6
We wish to thank the reviewer for pointing this out. We have corrected Table 1 and Figure 6 indexes.
Round 2
Reviewer 2 Report
The author of this paper addressed some of my concerns and highlighted.
For this, i appreciate his/her efforts.
Along with that, there are other concerns that need more consideration, such as open research directions related to Fordyce spots, which probably helps the readers/researchers after spending time in it (it is also helps other to find something new in this domain).
Thanks
Author Response
Dear Reviewer. Thank you for your advice. We have added a short section in discussion on open research areas:
“Further studies on the exact chemical compounds responsible UVFD luminescence and on the application of UVFD in other neoplastic and non-neoplastic dermatoses may contribute to wider use of this novel method.”
Reviewer 3 Report
My position remains the same, as the sample is so small I do not consider that you can use descriptive statistics, therefore you should sample for much longer and demonstrate soundness in your research.
Author Response
We thank Reviewer 3 for this comment and agree that larger research group could would make the power of our result stronger. Thus, unable to modify the design of the study, we have acknowledged the study limitations in a separate paragraph:
“There are several limitations of the study. The study covers retrospective data from a single Central European site, with all the patients being Caucasian. Further, prospective studies including non-Caucasian papulation should shed more light on the UVFD presentation of common genital dermatoses in a skin of colour. Our research group and control groups are innumerable, thus the distribution of patterns might be different in a real-life setting. Nevertheless, this study remains the largest study on dermatoscopy of FS published.”
Reviewer 4 Report
This authors assessed the ultraviolet reflectance dermatoscopy (UVRD) clues of Fordyce spots and their common clinical simulants: molluscum contagiosum, penile pearly papules, human papillomavirus warts, genital lichen planus, and genital porokeratosis. The authors analyzed 12 FS patients and 14 patients in the control group. Results show that UVRD is a fast, easy to apply, and low-cost modality that can further increase the diagnostic confidence and rule out selected infectious and non-infectious differential diagnoses if added to traditional dermatoscopic diagnosis.
Author Response
Thank you for your support.